# Assessing the Vulnerability and Adaptation Needs of Mozambique’s Health Sector to Climate: A Comprehensive Study

**DOI:** 10.3390/ijerph21050532

**Published:** 2024-04-25

**Authors:** Rachid Muleia, Genito Maúre, Américo José, Plácida Maholela, Isaac Akpor Adjei, Md. Rezaul Karim, Sónia Trigo, Waltaji Kutane, Osvaldo Inlamea, Lawrence N. Kazembe, Tatiana Marrufo

**Affiliations:** 1Department of Mathematics and Informatics, Universidade Eduardo Mondlane, Maputo 0101-11, Mozambique; 2Department of Physics, Universidade Eduardo Mondlane, Maputo 0101-11, Mozambique; genito.maure@uem.ac.mz; 3Department of Health Observation, National Institute of Health, Maputo 0205-02, Mozambique; americo.jose@ins.gov.mz (A.J.); placida.maholela@ins.gov.mz (P.M.); osvaldo.inlamea@ins.gov.mz (O.I.); tatiana.marrufo@ins.gov.mz (T.M.); 4Department of Statistics and Actuarial Science, Kwame Nkrumah University of Science and Technology, Kumasi AK-869, Ghana; isaac.adjei@gmail.com; 5Department of Statistics, Jahangirnagar University, Dhaka 1342, Bangladesh; rezaul@juniv.edu; 6World Health Organization Country Office, Maputo 280, Mozambique; casimirotrigos@who.int (S.T.); kutanew@who.int (W.K.); 7Department of Computing, Mathematical and Statistical Sciences, University of Namibia, Private Bag, Windhoek 13301, Namibia; lkazembe@unam.na

**Keywords:** climate change, vulnerability, adaptive capacity, health vulnerability index, Mozambique

## Abstract

Climate change poses severe consequences, particularly in sub-Saharan Africa, where poverty rates may escalate by 2050 without significant climate and development action. The health impacts are diverse, encompassing communicable and non-communicable diseases. Mozambique, a climate-vulnerable nation, has experienced significant natural disasters in the past 42 years, impacting its health system. This study aims to assess Mozambique’s health sector’s vulnerability and adaptation needs to climate change. Following a methodology proposed by the World Health Organization and the Intergovernmental Panel for Climate Change, a six-step vulnerability and adaptation assessment was conducted to conduct the Health Vulnerability Index (HVI) for Mozambique’s regions (n=161). The HVI integrates historical climate, epidemiological, and socio-economic data at the district level, and was computed using exposure, sensitivity, and adaptive capacity dimensions. The results revealed spatial patterns in exposure to climate variables, extreme weather events, and variations in sensitivity and adaptive capacity across the country. The HVI mirrored the exposure findings. Notably, high vulnerability was observed in several districts, while major urban centers displayed lower vulnerability. These findings highlight the country’s vulnerability to climate change and underscore the potential for adverse impacts on livelihoods, the economy, and human health. The study provides a foundation for developing strategies and adaptation actions.

## 1. Introduction

There is a growing body of evidence that Earth is warming at an unprecedented rate as a result of anthropogenic activities, such as deforestation, urbanization, population growth, industrialization, and the release of greenhouse gases [1]. The Intergovernmental Panel on Climate Change (IPCC) predicts an increase of 1.5–5.8 °C in global average temperature by 2100 as a result of greenhouse emissions [2]. Despite the fact that the wealthiest countries are disproportionately responsible for more emissions than developing countries, the consequences of climate change are more severe in the poorest countries, particularly in sub-Saharan Africa. In fact, developing countries have the least adaptive capacity justified by limited access to climate information, financial, human, and natural resources, as well as past and ongoing armed conflicts [3]. As a consequence of climate change, it is estimated that more than 30 million people in sub-Saharan Africa will fall into poverty if no substantial climate and development action is taken by 2050 [4]. Furthermore, it is expected that the gross domestic product will decrease by 3%, a scenario that poses a significant challenge to climate adaptation and resilience efforts, as it will lead to an increase in the number of people affected [5].

The effects of Climate Change on disease prevalence and human health are multifaceted. Studies show that Climate Change is behind the unprecedented increase in a myriad of non-communicable [6] and communicable diseases [7,8], such as vector and waterborne diseases, respiratory, and cardiovascular diseases [9,10]. For instance, rainfall affects water-borne diseases, and a large number of diarrhea pathogens are water-borne and will therefore be dispersed by Climate Change through water availability and temperature. Moreover, Climate Change can potentially trigger heat waves, floods, droughts, and storms, leading to high fatalities and injuries and altering disease scenarios [11]. Additionally, these events have other negative health consequences, such as depriving the health facilities of water, sanitation systems, and electricity (thus interrupting the cold chain of vaccines, reagents, and some drugs), blocking access routes to health facilities, destroying warehouses and stores of sanitary consumables, causing delays and absences in health personnel, etc.

Mozambique is considered to be at a high risk of being affected by Climate Change due to its geographical position along the coastline and downstream of main rivers in southern Africa. As a matter of fact, among the most vulnerable countries to natural disasters, the country ranks third [12]. In the last 42 years, the country has registered 15 droughts, 20 floods and 26 tropical cyclones [13], with these events often occurring consecutively within the same year. For example, in March 2019, the country was hit by the deadliest tropical cyclone to ever hit Africa (cyclone Idai), ranked as the second-deadliest on record [14,15]. The tropical cyclone caused significant flooding in Madagascar, Mozambique, Malawi, and Zimbabwe, killing more than 1500 people. In Mozambique, the cyclone affected mainly the provinces of Inhambane, Manica, Sofala, and Zambézia, leading to 1.85 million people in need of humanitarian assistance and protection [16]. In the province of Sofala, which was the most affected, the cyclone prompted a cholera outbreak, with more than 2000 cholera cases [16]. Subsequently, in less than two months, the country was again struck by another intense tropical cyclone named Cyclone Kenneth. Tropical Cyclone Kenneth affected the northern provinces of the country, Cabo Delgado and Nampula, displacing more than 18,000 people and causing 45 deaths [17,18].

The severe impacts of anthropogenic climate change on socio-economic conditions and health outcomes, particularly in developing regions such as sub-Saharan Africa. These impacts include potential increases in poverty rates, a decrease in Gross Domestic Product, and a rise in communicable and non-communicable diseases. Mozambique is exemplified as a highly vulnerable nation due to its history of natural disasters linked to climate change. The objective of the study is to assess the vulnerability and adaptation needs of the health sector to climate change in Mozambique, an essential step towards the creation of effective policies aimed at strengthening health systems, mitigating climate change impacts, and enhancing overall resilience.

## 2. Materials and Methods

### 2.1. Study Area and Climate

Mozambique is situated on the southeastern coast of the African continent between the latitudes 10°27′ S and 26°57′ S and longitudes 30°12′ E and 40°51′ E (see Figure 1), along the inter-tropic convergence zone responsible for Southern Africa’s rainfall patterns. To the north, the country is bordered by Tanzania, and to the west are Malawi, Zambia, Zimbabwe, South Africa, and Eswatini. The country covers an area of 801,590 ${\mathrm{km}}^{2}$ of which 13,000 km^2^ is water. The country is comprised of ten agroecological zones distributed throughout the 11 provinces of the countries [19]. Furthermore, the country has about 2470 km of coastline along the Mozambique Channel and the Indian Ocean, spanning 10 of its provinces. The geographic location of Mozambique along the coast in the Indian Ocean makes the country vulnerable to tropical cyclones, floods, and droughts [20,21].

The climatic conditions in Mozambique vary from tropical to subtropical. The country has two seasons, a cool and dry season–which starts from May to September–, and a hot-humid season, between October and April. Precipitation is more abundant in the Center and Northern regions of the country, with values ranging from 800 to 1200 mm per year. The southern region of the country is generally drier inland than on the coast, where annual rainfall reaches around 800 mm and decreases to about 300 mm. The average air temperatures, in general, vary between 25 °C and 27 °C in the summer and 20 °C and 23 °C in the winter [22].

Furthermore, following the climatic classification of Köppen–Geiger, it is possible to find four predominant climatic regimes: tropical rain savannah, dry savannah, dry desert climate, and humid temperate climate. The tropical rain savanna climate occurs mainly in the northern and coastal regions of the country. In contrast, the dry savannah climate occurs in the inland parts of the central and southern sedimentary terrains. The dry desert climate is not so abundant, occurring in a small area around the Limpopo River in the province of Gaza. The humid temperate climate can be found in the upland areas of Gurué, Manica, Angónia and Lichinga [22,23].

### 2.2. Study Design

The vulnerability and adaptation assessment (VAA) study was conducted following a methodology proposed by the World Health Organization [24], which is in line with the IPCC’s fourth assessment report. As recommended by the IPCC and the vulnerability sourcebook, in this VAA, we considered an integrated approach, where this VA took into account the climate-sensitive health aspects, social, economic, and environmental aspects of vulnerability [25]. In accordance with WHO [24] recommendations, the VAA was carried out following six steps: (1) define the scope of the VAA—at this stage, the objectives of the VAA were specified, the geographic area (for this assessment it was decided to work at district level for the whole country), review of relevant policies, stakeholder engagement for disease/health priorities (this involved workshops, drawing experts from different organizations, including public health disease surveillance, epidemiologist, climatologist, etc.); (2) Identification and selection of indicators—this included defining the necessary data to be collected as well as mapping the existing data sources; (3) Obtaining data—this phase consisted of gathering all relevant data for the VAA (climatological, sociodemographic, health, and economic data), data cleaning and formatting; (4) vulnerability analysis—at this stage the HVI was defined and quantified, preceded by normalization of all the indicators; (5) results validation—the results validation process was accomplished by discussing the computed HVI with Ministry of Health, different stakeholders and experts; and the sixth step involved the dissemination of the results. Additionally, the HVI computed was compared using the approach defined herein to computations through principal component analysis. The VAA ended with a final report that included recommendations for health sector adaptation and climate change monitoring and technical presentation to stakeholders.

### 2.3. Data Source

#### 2.3.1. Climate Data

To evaluate the contribution of climate variables in the HVI, historical monthly data on temperature, rainfall, and relative humidity from 1970 to 2016 were used. Historical monthly data were used to compute the average annual variation. The data were obtained from the National Institute of Meteorology (INAM), which provides data collected from weather stations. Additionally, due to the poor geographic coverage of national weather stations, the data from INAM was complemented by satellite data. Data on temperature were extracted from publicly available National Oceanic and Atmospheric Administration (NOAA) at a spatial resolution of 0.5, available at https://www.esrl.noaa.gov/psd/ (accessed on 20 December 2019). Monthly rainfall data were obtained from Multi-Source Weighted-Ensemble Precipitation (MSWEP) version 2.1 (http://www.gloh2o.org/, accessed on 20 December 2019), which is based on ground interpolation data obtained from the European Centre for Medium-Range Weather Forecasts (ECMWF) ERA5 [26]. Similarly, data on relative humidity (RH), with a spatial resolution of 0.25 degree, was obtained from the ERA5. To compute the annual changes, the monthly climate data were averaged over a year.

Data on extreme event data, such as floods, cyclones, and droughts were extracted from DesInventar, available at http://www.desinventar.net (accessed on 26 July 2023). DesInventar is a methodological tool for generating National Disaster Inventories and constructing databases of damage, losses and, in general, the causes and impacts of disasters. This tool provides data on losses caused by disasters associated with natural hazards. Further details on the DesInventar can be found in [27]. From this tool, we were able to extract data on disasters from 1979 to 2012. The current study considered data from 1980 to 2019. To fill the gap between data from 2012 and 2019, data from administrative sources were considered.

#### 2.3.2. Epidemiological Data

To compute HVI, epidemiological data at the district level on annual cases of malaria, diarrhea, tuberculosis, HIV, and chronic and acute malnutrition, were collected. The data comprised a period of 2017 and 2018, where the cumulative cases for this period were used. These data were obtained from the National Directorate for Public Health in the Ministry of Health.

#### 2.3.3. Socio-Economic Data

Table 1 provides the list of all indicators used to conduct the vulnerability and adaptation assessment. For the current study, socio-economic data were considered that were collected from various sources: the Ministry of Health (MISAU), the Technical Secretariat for Food Security and Nutrition (SETSAN), and the National Institute of Statistics (INE). From the latter, demographic data, literacy rate, per capita expenditure, and the rate of accessibility to water, sanitation, and hygiene (WASH), were obtained.

### 2.4. Data Analysis

Prior to the computation of the HVI, all the indicators were normalized. Normalization was applied to allow comparability of the HVI across the districts and to remove the effect of scale of different indicators on the computation of the HVI. The normalization was accomplished using the following mathematical equation:(1)$$Y_{i}=\frac{X_{i}-X_{\max}}{X_{\max}-X_{\min}},$$
where $X_{i}$ represents the value for a given indicator for the *i*th district, and $X_{\min}$ and $X_{\max}$ represent the minimum and maximum values of the indicator, respectively. Note that the normalization through Equation (Equation 1) ensures that the values are bounded between 0 and 1.

#### 2.4.1. Vulnerability Assessment Model

Following Shah et al. [28] and Luh et al. [29], the HVI, at the district level, was defined as follows:(2)HVI=(E−AC)×S,
where *E* represents the exposure index, *S* the sensitivity index, and AC the adaptive capacity index. The exposure index was defined using the weighted sum method, also known as the additive or direct summation approach [30,31]. This approach builds as follows:(3)$$E_{j}=\sum_{i=1}^{N}w_{i}Y_{i,j},$$
where $Y_{i,j}$ denotes the normalized indicator at district *j* and $w_{j}$ is the weight assigned to $Y_{i}$, such that $\sum_{i=1}^{N}w_{i}=1$, and *N* represents the total number of indicators. For the exposure component, *N* is taken to be four. For the purpose of this study, equal weights for all the normalized indicators were considered, which correspond to the arithmetic mean. Similarly, the sensitivity index was computed as an average score from a set of 12 indicators (the indicators used for deriving the sensitivity index are listed in Table 1). The derivation of the adaptive capacity index considered a set of 16 indicators, which are also presented in Table 1. Likewise, the exposure index and sensitivity index, as well as the adaptive capacity, were computed as an average score, where all the indicators were assigned the same weight. After deriving all three sub-indices (exposure index, sensitivity index, and adaptive capacity index), the HVI was calculated for a total of 161 districts using Equation (Equation 1). The HVI, including the sub-indices, were then classified into quintiles, creating five categories of classification.

It is important to mention that some indicators, such as per capita expenditure, lacked information at the district level. Thus, to derive the HVI for such districts, information at the provincial level was used by assigning the same information to the districts in the same province. In cases where the district lacked information for a particular indicator, a value corresponding to the arithmetic mean of the observed values for the other districts of the same province was imputed. All the analyses herein were performed using the R software for statistical computing and graphics, version 4.0.3 [32].

**Table 1 ijerph-21-00532-t001:** List of indicators used to compute the exposure, sensitivity, and adaptive capacity index.

Vulnerability Determinant	Components	Indicators	Data Source
	Changes on rainfall, temperature and RH	Average temperature variation (1970–2016)	NOAA
Exposure		Average rainfall variation (1970–2016)	MSWEP
		Average variation in the RH (1970–2016)	ECMWF
	Extreme events	Historical data on the frequency of floods, droughts and cyclones (1979–2019)	DesInventar data [27].
Sensitivity	Natural capital (Ecosystem/Geography—Risks)	Frequency of cholera outbreaks (2014–2019)	MISAU
		Frequency of food insecurity episodes (2016–2019)	
	Natural capital (Demography and vulnerable population)	Population density (2017)	
		Percent of children under five in the district (2017)	
		Percent of children aged 5–15 years in the district (2017)	Census data [33]
		Percent of women in the district (2017)	
		Percent of elderly people (over 60 years old) in the district (2017)	
	Vulnerable population due to health conditions	HIV positivity rate (2017–2018)	
		Rate of reported cases of tuberculosis (2017–2018)	
		Average number of cases of acute and chronic malnutrition per 100,000 inhabitants (2017–2018)	MISAU (National Directorate for Public Health)
		Average number of cases of malaria per 100 inhabitants (2017–2018)	
		Average number of reported cases of diarrheal diseases per 100,000 inhabitants (2017–2018)	
	Financial resources	Per capita public sector health expenditure (2018)	MISAU (division of administration and finance)
	Health services	Ratio of the total number of inhabitants to the total number of health units in the district (2018)	Census data [33] and SARA report [34]
		Percentage of population living within the coverage radius of a health facility (2018)	
	Human resources	Ratio of medical workers per 100,000 inhabitants	MISAU (division of human resource)
Adaptive capacity		Ratio of nursing workers per 100,000 inhabitants	
		Ratio of workers in the midwifery area per 100,000 inhabitants (2018)	
		Number of inhabitants per health elementary multipurpose agents (2019)	MISAU (National Directorate for Public Health) and census data [33]
	Water and sanitation	Percentage of population with access to safe water sources (2017)	Census data [33]
		Percentage of population with access to safe latrines (2017)	
	Social capital (Social determinants of health)	Percentage of literate population, men (2017)	Census data [33]
		Percentage of population with primary education, men (2017)	
		Percentage of population with secondary education, men (2017)	
		Percentage of literate population, women (2017)	
		Percentage of population with primary education, women (2017)	
		Percentage of population with secondary education, women (2017)	
		Per capita expenditure (2014)	National household budget survey report [35]

## 3. Results

### 3.1. Exposure

Exposure scores were calculated considering information on climate variation (temperature, rainfall, and relative humidity) and extreme events. Figure 2 shows the geographic distribution of average temperature variation, rainfall variation, and relative humidity variation. The map on average temperature variation highlights two differentiated regions, one with negative variation (suggesting a decrease in the temperature) and another with a positive variation (indicating an increase in the temperature). From the map, it is possible to observe that the heating is more evident in the southern part of the country (Maputo and Gaza province) and in the coastal provinces of the country (Inhambane, Sofala, Zambézia and Cabo Delgado). Additionally, we observe that more inland districts present a reduction in the temperature, as is the case of all the districts of Tete province and the majority of districts of Niassa and Manica province. It is worth noting that districts in the coastal areas of Inhambane, Sofala, and Zambézia also experience a reduction in temperature.

With regard to the average rainfall variation, as illustrated in Figure 2b, it can be observed that the vast majority of the districts, from 1979 to 2016, experienced an increase in the amount of rainfall. Furthermore, it is observed that the increase in the amount of rainfall is substantial in some districts located on the coast of Inhambane and Gaza province (Inhambane, Maxixe, Jangamo, Inharrime, Mandlakazi e Bilene) and some inland districts of Zambézia province (Mulombo, Gurué, and Alto-Molocue). Other districts with a sharp increase in the rainfall amount are the districts of Chifunde in the northern part of Tete province, and Macanhelas and Marrupa in Niassa. Amongst all the 161 districts, only 11 districts are located in Zambézia, in the center of Manica province, and in Sofala province.

Concerning the relative humidity (Figure 2c), the country registered a decrease in the relative humidity (RH), which varied between −0.42% and −4.59%. From Figure 2c, it can be noted that the reduction in the RH is not substantial across the country. Nevertheless, it is apparent that it tends to decrease sharply as we move away from the coast. This trend is more evident in the northern and central parts of the country, mainly in the provinces of Nampula, Niassa, and Zambézia. It is also observed that in Tete province the changes in RH are negligible.

Figure 3 shows the incidence of extreme events—cyclones, droughts, and floods—in Mozambique for each of the districts in the last 40 years. From this figure, it can be observed that tropical cyclones (Figure 3a) are not evenly distributed across the country, affecting districts along the coast the most. The map also suggests that Mozambique has high exposure to tropical cyclones. Additionally, it is possible to observe that tropical cyclones are more frequent in the Nampula province (in the districts of Moma, Larde, and Angoche), Inhambane province (Massinga), Sofala province, Zambézia province, and Maputo province.

Figure 3b shows the spatial distribution of the occurrence of droughts in Mozambique. From the map, it can be noted that in the last 40 years, 72 districts have been affected by droughts. Furthermore, it is apparent that most often, the droughts affect provinces from the southern region of the country, with particular emphasis on districts from Inhambane and Gaza provinces. The phenomenon also affects, with less frequency, some districts from Maputo, Manica, Sofala, Tete, Nampula (districts located along the coast), and Cabo Delgado province.

With regard to floods, the map in Figure 3c shows that, although the frequency of occurrence of floods is not evenly distributed across the country, almost all the districts were once affected by floods in the last 40 years. From the map, one can observe that floods are more frequent in the main river basins in the center (Zambezi, Púngue, and Buzi basins) and in the south of the country (Save, Limpopo, Incomáti, Umbelúzi, and Maputo basins), affecting districts across Zambézia, Tete, Sofala, Manica, Gaza and Maputo provinces. In the country’s northern region, despite the low frequency in general, it can be observed that the Messalo River basin in Cabo Delgado province is also affected with some frequency by floods.

### 3.2. Exposure, Sensitivity, and Adaptive Capacity Index

Figure 4 shows the exposure, sensitivity, and adaptive capacity index calculated for 161 districts. The exposure and sensitivity maps are displayed in five intervals obtained using the quintiles, where the yellow and the dark red correspond to the lowest and highest exposure (sensitivity), respectively. In contrast, for the adaptive capacity map, the yellow and the dark red represent the highest and lowest adaptive capacity, with purple representing the presence of a critical situation. From Figure 4a, it is noted that exposure to climatic hazards is considerably higher in the southern region of Mozambique, especially in districts located on the coastline of Inhambane province. The map also shows that exposure is higher in some districts of Sofala, Zambézia, and Nampula province, alongside the Zambeze watershed. Cabo Delgado and Niassa province, in the northern region of Mozambique, appear to have lower exposure to climate change.

With regard to sensitivity to climate change, Figure 4b exhibits a spatial pattern, where low sensitivity to climatic hazard is vastly observed in the northern region of Mozambique, whereas moderate sensitivity is observed in the central and southern region, mainly in Gaza, Inhambane, and Sofala province. Adaptive capacity, which reflects the district’s ability to cope with climatic hazards, is presented in Figure 4c. Overall, the adaptive capacity (AC) across the country is very low. Despite the apparent homogeneity of AC, it is evident that the southern region of Mozambique has a greater AC index when compared to the central and northern regions of Mozambique. Moreover, the map shows that the AC is higher in urban areas, which coincides with the capital provinces and contrasts with rural areas. The AC index was also visualized on a map excluding the extreme values corresponding to the main capital cities (Figure 4d). It can be noted that after excluding the urban districts, a clear pattern emerges, where the southern region appears to have high AC and the northern region has low AC. Furthermore, it is observed that Maputo province, which is located in the southern part of Mozambique, has all of its districts with high AC.

### 3.3. Adaptive Capacity Determinants

For a better understanding of the adaptive capacity index and to investigate which of the determinants contribute the most in the resilience of a particular district, we analyzed the sub-indices of each of the determinants for the adaptive capacity components, namely, access to health services, human resources for the medical sector, wash and sanitation and financial resources for the medical sector. Likewise, the exposure, sensitivity, and adaptive capacity index, the sub-indices for each of the adaptive capacity components’ determinants, was computed following the approach described in Section 2.4.1.

#### 3.3.1. Access to Health Services

Figure A1a,b map the “access to health services” index. In this figure, the mapped values consider the districts (Figure A1a) or exclude the districts with extreme values (Figure A1b), indicated as grey areas. When we consider all districts it is observed that only urban districts in Maputo, Matola, Chimoio, Tete and Nampula appear to have high access to health services. We also observe that the vast majority of the country is characterized by districts with very low access to health services. Looking at Figure A1b, the scenario that excludes districts with extreme values, we observe that, contrary to the previous scenario, the health access index shows some degree of heterogeneity. Furthermore, some clustering of districts with very low access to health can be observed: some located in the southern region of the country (in the provinces of Gaza and Inhambane) and others in the northern region of the country (in the province of Niassa). Moreover, in the southern region, it is possible to see that almost all districts of Gaza province have a very low access to health services. The map (Figure A1) also reveals that most districts from Nampula province have either moderate or high access to health services.

#### 3.3.2. Human Resources for Health Services

We further assessed the distribution of the human resources in the health sector. For this purpose, we considered a human resource index, which considers all the human resource indicators listed in Table 1. The human resource index is mapped in Figure A1c,d. Likewise other indices, we present two scenarios, one showing the human resource index (HRI) for all districts (Figure A1c,d). From Figure A1c, one can see that the HRI is homogeneous across the country, with most of the districts exhibiting a very low HRI, except Inhamabane, Pemba, Lichinga, and Chicualacuala districts, which lie into the moderate category. Nevertheless, when we discard districts with extreme values, we note that the HRI appears to be heterogeneous across the country. Furthermore, it can be noted that districts with high HRI are concentrated in the southern region of the country, whereas districts with low HRI are focused on the central and northern regions of the country.

#### 3.3.3. Water and Sanitation

Figure A1e,f show the country’s water and sanitation index (WSI). Similarly, as with other components, we analyze the water and sanitation component by considering all the districts, as shown in Figure A1e, and discarding districts with extreme values for WSI (Figure A1f). One can see that when all the districts are considered, the vast majority of the districts exhibit either low or very low WSI. However, we note that even discarding districts with the highest WSI, the distribution of WSI across the country continues to be critical, where we note that a total of 125 districts, which corresponds to 92% of the country’s territory and 71.5% of the total population, has a low WSI. Additionally, it can be noted that almost the entire region in the north has poor access to safe water and sanitation. For instance, we note that in Zambézia province, every district has a very low WSI, except Quelimane. Contrary to what is observed in the country’s northern region, a substantial part of the southern region, mainly in Maputo province, appears to have a high WSI.

### 3.4. Health Vulnerability Index

Figure 5 shows the districts’ overall health vulnerability index maps. The maps show that, in general, the HVI is high all over the country, indicating that the health sector is highly vulnerable to climate change. Furthermore, an apparent inequality between urban and rural districts can be observed, whereby urban districts appear to have a lower HVI. The map (Figure 5a) also suggests that districts from the northern region in Niassa and Cabo Delgado province are less vulnerable than the majority of the districts in the southern region of Mozambique. The HIV was also visualized discarding urban districts that coincide with provincial capitals, allowing one to have a clear picture of districts that are, in fact, vulnerable to climate change. The map that shows the HVI discarding urban districts with extreme values is shown in Figure 5b. From this map, it can be noted that only eight districts have a very high HVI and, contrary to the scenario in Figure 5a—where the majority of the country was characterized by districts with high vulnerability index—the majority of the districts fall into moderate HVI. Moreover, it is observed that Cabo Delgado province, despite having a vast coastal area and a lower adaptive capacity, has lower HVI than many districts in Mozambique’s southern and central regions.

### 3.5. Vulnerability Index to Droughts, Floods and Cyclones

In the current study, we also computed the HVI for each extreme event, drought, flood, and cyclone. This index allows one to identify more vulnerable districts to a particular, extreme event. The HVI to particular, extreme events of droughts, floods, and cyclones are presented in Figure 6a, Figure 6b and Figure 6c, respectively. This figure shows that the southern region of Mozambique, emphasizing the provinces of Inhambane and Gaza, is highly vulnerable to droughts. The map in Figure 6a also shows that Cabo Delgado province in the northern region of Mozambique is more vulnerable to droughts among provinces located in this region, particularly districts of Mueda and Mossuril. The geographic distribution of the HVI to floods is presented in Figure 6b. From this figure, we can observe that the geographic distribution of incidence of floods in Figure 3c, the geographic distribution of districts highly vulnerable to floods, follows the main watersheds in the country. Furthermore, we observe that among districts with a high vulnerability index, four are located in Zambézia province, three in Tete and Sofala province, and two in Gaza province. Overall, the Sofala province appears to be much more vulnerable to floods than other provinces. With regard to vulnerability to cyclones, Figure 6c suggests that districts that are much more vulnerable to cyclones are those located in the coastal area of the country. Additionally, we observe that in Nampula and Sofala province, not only districts situated on the coast appear to be vulnerable to cyclones but also inland districts.

## 4. Discussion

Vulnerability is a cross-cutting multidisciplinary research theme. As a result, it can be perceived differently depending on the field to which it is applied. For instance, Adger et al. [36] define vulnerability to climate change as the propensity of human and ecological systems to suffer harm and their ability to respond to stresses imposed by climate change effects. One study Schröter et al. [37] refers to vulnerability to climate change as the likelihood that a specific coupled human-environment system will suffer harm as a result of exposure to stresses associated with societal and environmental changes, taking into account the adaptation process. Some authors Rao et al. [38], who look into vulnerability in the context of agriculture, define vulnerability as the propensity of an organism to experience climate shocks and suffer loss in production and/or income from agriculture. In this paper, we rely on the definition of the Intergovernmental Panel on Climate Change, which is defined as “the extent to which a system is susceptible to or unable to deal with the effects of climate change, including extreme weather events” [39]. Thus, following this definition, vulnerability can be seen as a function of three dimensions, namely, exposure, sensitivity, and adaptive capacity. Exposure refers to the extent of climate stress on a particular unit of analysis, while sensitivity can be understood as the extent to which climate-related stimuli have an effect on a system, either favorably or unfavorably [40]. The sensitivity is determined by both socio-economic and ecological factors and measures the extent to which a particular group will be affected by environmental stress [41]. The adaptive capacity, as defined by the IPCC, can be understood as the ability of a system to respond to the effects of climate change.

To assess the vulnerability, the framework recommended by the IPCC and WHO was considered, whereby the indicators collected for the assessment were classified into three dimensions, namely: exposure, sensitivity, and adaptive capacity, allowing each dimension to contribute to the computation of the HVI [24]. The indicator method is well-documented in the literature [42,43,44]. This approach is highly recommended among scholars as it allows one to compute the vulnerability index for the following reasons: the vulnerability index can be computed at any scale—household, district, province, and national level—allowing one to monitor the trend over time; the approach is multi-dimensional, which means it can capture multiple dimensions of vulnerability; and the approach is appropriate for identifying most vulnerable places [29,45].

To the best of our knowledge, this is the first study to assess the vulnerability of Mozambique’s health sector to climate change on a national scale. Our results pointed out that exposure to climate change is substantially high in the southern region of Mozambique. Additionally, the results showed that coastal districts located in other regions of the country are also highly exposed to climate hazards, emphasizing Nhamatanda, Buzi, Beira, and Marromeu districts in Sofala province. The elevated exposure index among districts located in the coastal area can be justified because this region of the country is exposed to several coastal hazards, including tropical cyclones, storm surges, and sea level rise [46]. In fact, Mozambique has a vast coastline of about 2500 km in an area highly prone to tropical depressions and typhoons. In the last 40 years, Mozambique has been hit by about 21 cyclones, most of them affecting coastal districts. These results are consistent with the study by Cabral et al. [47] on Mozambique’s exposure to coastal climate hazards and erosion. The authors found that districts on the central coast had more than 10% of their coastal areas with higher exposure to climate hazards.

The results also revealed that the southern region of Mozambique is highly exposed to climatic hazards. In fact, it is worth noting that this part of the country is located in one of the largest river basins of the Southern African development community (SADC). The Limpopo River basin has been subject to several floods of great magnitude. In the last 70 years, in 1955, 1967, 1972, 1975, 1977, 1981, 2000, and 2013, severe floods have affected the Limpopo river basin, caused mainly by heavy localized rainfall, tropical cyclones, and inadequate management of upstream dams and wetlands [48,49]. The analysis also showed that contrary to other regions that only experience one time of climatic hazard, the southern region experiences all the climatic hazards, which could be behind the elevated exposure index in this part of the country. The results also indicated that the districts along the Zambezi River basins (the second largest river basin in the African continent) in Zambézia and Nampula province are also susceptible to extreme weather events, mainly to floods.

With regard to the sensitivity of the country’s population to climate change, the results suggest that despite the country being highly exposed to extreme weather events, the country is vastly characterized by districts with either low or moderate sensitivity. The reasons behind the generalized low sensitivity index nationwide are not evident. Nevertheless, a more in-depth analysis suggests that although the country is prone to extreme weather events, few cholera outbreaks were experienced. Districts located in Mozambique’s central and northern regions, such as Beira, Caia, Tete, Nampula Melucu, and Mecufi, which have frequent cholera outbreaks, are the ones that stood out with high sensitivity. In fact, Mozambique’s central and northern regions are more prone to floods than the southern region, which often cause people to move and settle into overcrowded places with poor sanitation conditions, increasing the risk of cholera outbreaks. For instance, in 2019, an intense tropical cyclone hit the central region of Mozambique, leaving the region entirely devastated, causing a cholera outbreak with more than six thousand cases [50,51]. Additionally, it is worth emphasizing that districts with high sensitivity also had a high risk of food insecurity between 2016 and 2019. With the northern region being prone to floods as a result of climate change, it is expected that there is a destruction of food crops, storehouses, and livestock, and as a result, a decline in food availability and an increase in the risk of food insecurity, which in turn would increase nutritional problems among children under five [52,53]. According to the Mozambique Technical Secretariat for Food Security and Nutrition (SETSAN), chronic food insecurity is most prominent in the central and northern regions of the country [54].

The results further revealed that Mozambique generally has a low adaptive capacity, being more pronounced in the central and northern regions of the country. Poverty, limited investments in advanced technology, and the fragility of infrastructure and social services, particularly health and sanitation, may explain the country’s poor adaptive capacity to climate change [55]. As a matter of fact, it is expected that regions or districts with well-developed institutions coupled with higher levels of capital and stores of human knowledge are considered to have higher adaptive capacity [56]. For instance, data on the fourth national evaluation on poverty and wellness in Mozambique indicate that the northern region has the highest poverty rates, with the province of Niassa standing out with rates around 60%. Hence, the expected low adaptive capacity among districts from this province. Additionally, a critical examination of the results revealed that the central and northern regions have poor sanitation conditions, which is more evident in rural districts. Add to that, the prevalence of water-borne diseases among children under five is more pronounced in the central and northern regions of Mozambique, varying between 4.1% and 14.5% [57].

Among the determinants contributing to high adaptive capacity, we also found that access to health services is still deficient in Mozambique, mainly in Niassa (northern region) and the provinces of Gaza and Inhambane (southern region). Nevertheless, it is worth emphasizing that the identified provinces have the least population density with sparse households, which at some point hinders the provision of health services and access to them. These findings align with a study by dos Anjos Luis and Cabral [58]. The authors analyzed the geographic accessibility to primary healthcare centers in Mozambique. They found that the provinces of Nampula (northern region), Zambézia and Tete (central region), and Inhambane (southern region) have the highest number of villages outside 60 min of a health facility. The results also indicated that access to healthcare is urban-biased. The inequalities in the distribution of health facilities can be justified by the fact that many infrastructures in the rural area were destroyed during the civil war, which makes many of the rural districts much more vulnerable to the impact of climate change [59]. The lack of access to healthcare not only puts the districts into an increased vulnerability but also affects the abilities of districts to cope with the effects of climate change, such as the rise of cases of cholera, diarrhea, malaria, and food insecurity, after the country has experienced an extreme weather event.

Although the study clarifies the spatial trend of the HVI in Mozambique, there are some limitations. Due to a lack of data on some indicators for some of the districts, proxy indicators were used to assess the exposure, sensitivity, and adaptive capacity. Therefore, one should consider the limitations associated with using proxies: under-representation or over-representation. Another major limitation was the lack of weather stations for all the country districts, which made it impossible to obtain in situ data. Additionally, for several reasons, the historical series of climatic data collected by the operational weather stations present numerous gaps, making it even more difficult to obtain complete and consistent databases to conduct an analysis of this nature. To overcome this situation, the study used reanalysis data. Nevertheless, climate data from reanalysis can be over- or under-estimated in specific periods and zones of the country [60]. For some indicators, it was not possible to obtain recent data. For instance, for health determinants and water and sanitation data, we had to rely on 2007 census data, despite the 2017 census having been carried out, as by the time of VAA the 2017 census data were not publicly available. Therefore, it is possible that the adaptive capacity index is somehow underestimated, as indicators for these components might have improved over the last 12 years.

It is noteworthy that the VAA report also included a case study on the situation of two emerging climate-sensitive diseases (dengue and chikungunya) in Mozambique and an assessment of the impact of climate change on malaria, watery diarrhea, and cholera. Nevertheless, for the current paper, we focused on the analysis of the HVI.

## 5. Conclusions

The research findings suggest that the country, barring its primary capital cities, is significantly vulnerable to climate change, a condition attributed to its geographic location within the inter-tropical convergence zone and downstream of shared watersheds. The country faces a high risk of severe weather-related disasters, including tropical cyclones, floods, and droughts [61]. This risk is particularly concerning considering the high dependency on subsistence farming, as exemplified by the population in Mozambique. Climate variations, therefore, have the potential to influence the livelihoods and the economy severely, often leading to widespread destruction. Recent cyclones, namely Idai and Kenneth, exemplify this vulnerability, causing significant devastation, including a nutritional crisis that affected about one million people [62]. The study acknowledges that immediate changes to the country’s climate change exposure or sensitivity are unrealistic, but it emphasizes the development of policies, strategies, and plans to bolster community adaptation capabilities. Given the multi-sectoral nature of the country’s vulnerability, a coordinated approach is required, and the findings from this study can guide authorities in creating targeted, district-specific policies. Especially in coastal districts, which are highly prone to tropical cyclones, the construction of resilient infrastructure is crucial. Finally, the study underlines the importance of women’s empowerment strategies, considering their high dependency on natural resources and the disproportionate impact of climate change on women, particularly those leading households in urban poverty.

## Figures and Tables

**Figure 1 ijerph-21-00532-f001:**
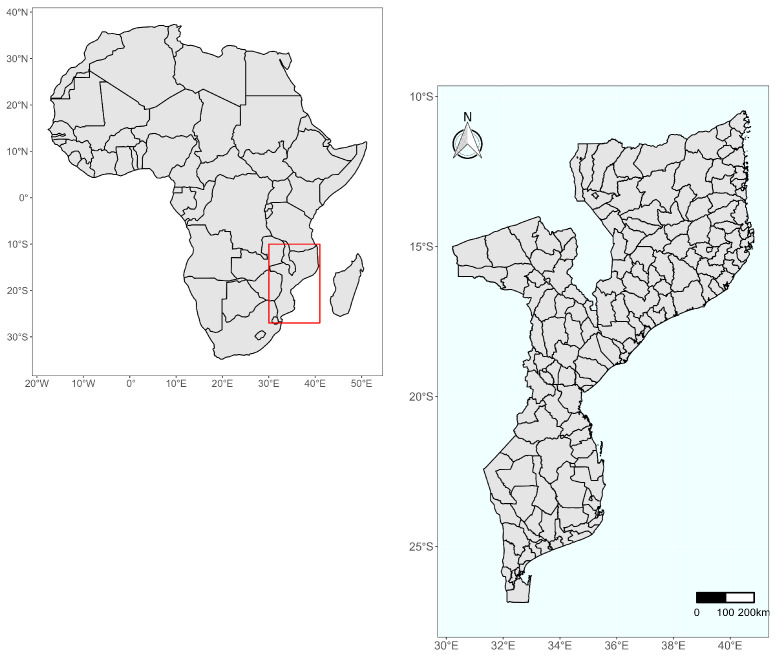
Map of the study area. On the left is the African continent, and marked with a red square is the study area of which the enlargement shows Mozambique and its 161 districts. In Appendix A, we provide a map with all the districts numbered and their corresponding names. The number and names of each district are presented Figure A2 and Table A1.

**Figure 2 ijerph-21-00532-f002:**
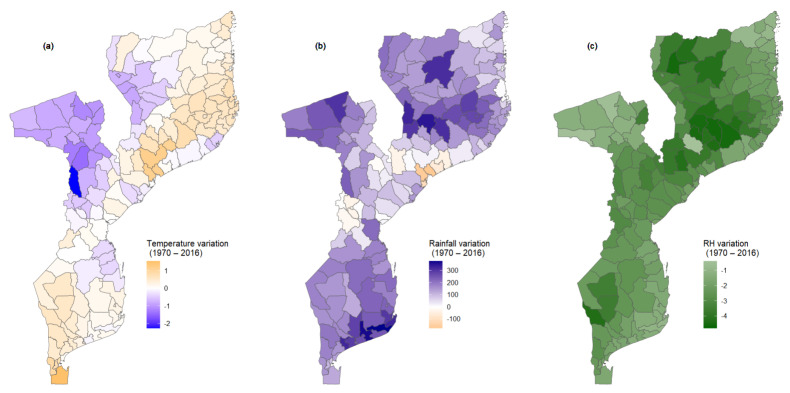
Spatial distribution on climate variable variation. (**a**) shows the spatial distribution of temperature variation. (**b**) shows the spatial distribution of rainfall variation and (**c**) shows the spatial distribution of relative humidity variation for a period between 1979 and 2016.

**Figure 3 ijerph-21-00532-f003:**
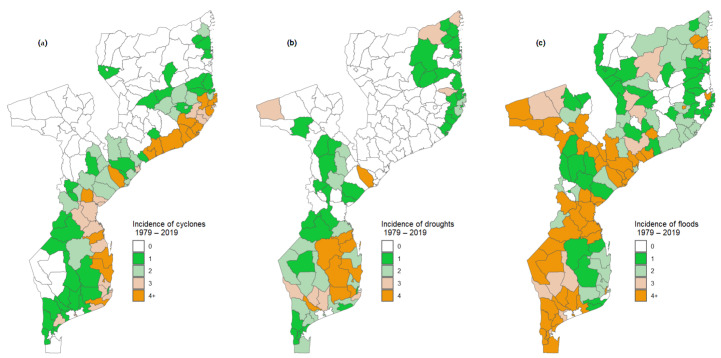
Spatial distribution of extreme events. Maps (**a**–**c**) show the incidence of cyclones, droughts and floods for each district across the country, respectively. Extreme events are reported between 1979 and 2019.

**Figure 4 ijerph-21-00532-f004:**
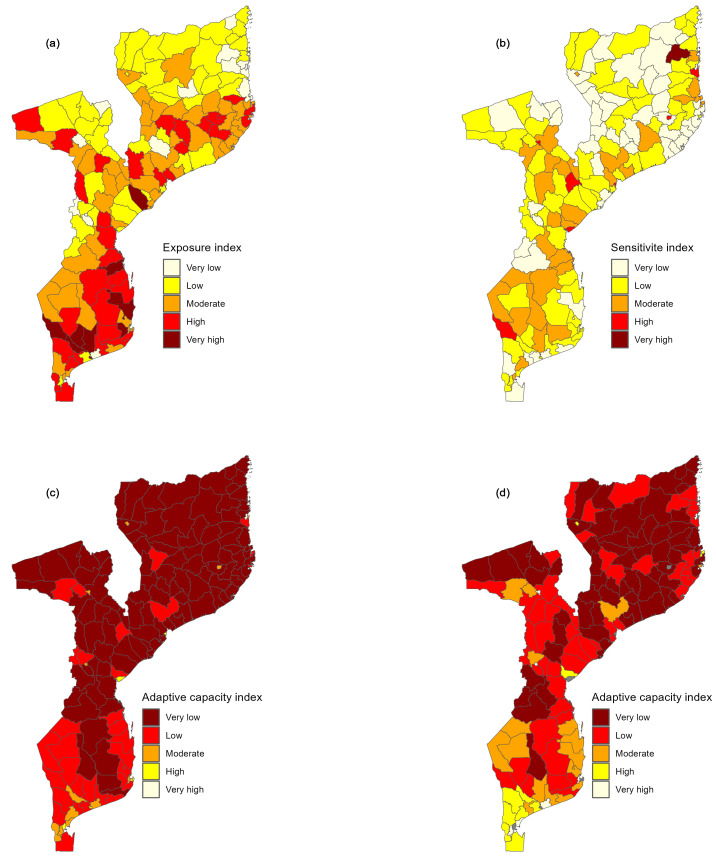
Spatial distribution of exposure (**a**), sensitivity (**b**), and adaptive capacity index (**c**) at the district level across the country. Map (**d**) shows the spatial distribution of adaptive capacity excluding extreme values, which coincide with the main capital cities.

**Figure 5 ijerph-21-00532-f005:**
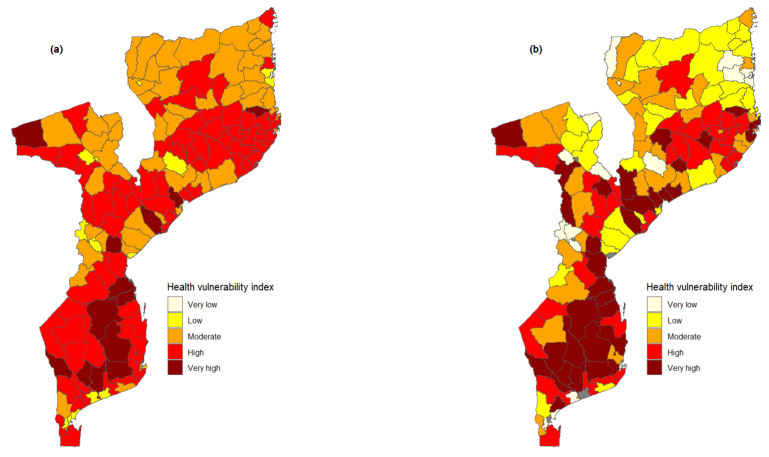
Maps of health vulnerability index by districts. Map (**a**) presents the health vulnerability index (HVI) without discarding extreme values, and Map (**b**) shows the HVI discarding extreme values indicated as grey areas.

**Figure 6 ijerph-21-00532-f006:**
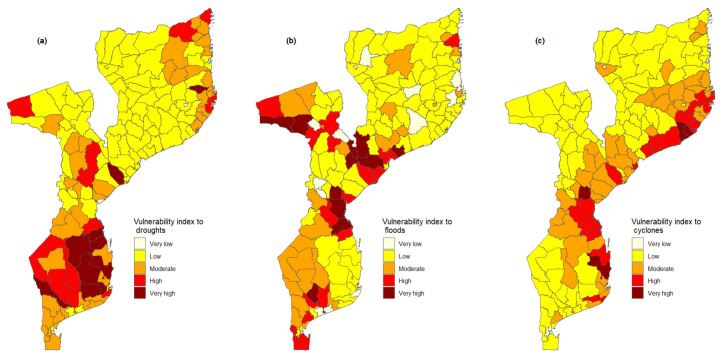
Spatial distribution of health vulnerability index (HVI) to specific hazards. Maps (**a**–**c**) show the HVI to droughts, floods, and cyclones, respectively. The HVI to specific hazards was computed for each of the 161 districts.

## Data Availability

Data used in this paper can be obtained from the author upon reasonable request.

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
