# Peer review of "Assessing the Vulnerability and Adaptation Needs of Mozambique’s Health Sector to Climate: A Comprehensive Study"

_ijerph, 2024, doi:10.3390/ijerph21050532_

Round 1
Reviewer 1 Report
1. Important component of the map are missing from the figure 1. (legend, north arrow, geographic coordinates, scale).
2. Mention the acronym on the first place when full form appears including its use in Abstract. Abstract is a different component of an article and therefore besides the mention in abstract, acronym must be mentioned with the first appearance of full form; e.g. CC and HVI are not mentioned anywhere in the introduction section. However, using CC is not suggested and instead use Climate Change.
3. Statement of the problem and objectives of the study must be mentioned in the last paragraph of the introduction section. Which are found missing.
4. Suggest references for citation
5. Replace “that is” with “where” in L 102.
6. Remove “Fritzsche et al.“ from L 104.
7. Rewrite the sentence L 119, here Terminate is not the right term to use.
8. Reframe the sentence L 146, don’t use elsewhere, mention the exact source.
9. Cite different research articles for the communicable (Ref i and ii) and non-communicable (Ref iii) diseases in L 35-37.
i. Parihar, R.S.; Bal, P.K.; Thapliyal, A.; Saini, A. Climate Change Projections and Its Impacts on Potential Malaria Transmission Dynamics in Uttarakhand. J. Commun. Dis. 2022, 54, 47–53, doi:10.24321/0019.5138.202249.
ii. Parihar, R.S.; Bal, P.K.; Saini, A.; Mishra, S.K.; Thapliyal, A. Potential Future Malaria Transmission in Odisha Due to Climate Change. Sci. Rep. 2022, 12, 9048, doi:10.1038/s41598-022-13166-5.
iii. Rother, H.-A. Controlling and Preventing Climate-Sensitive Noncommunicable Diseases in Urban Sub-Saharan Africa. Sci. Total Environ. 2020, 722, 137772, doi:10.1016/j.scitotenv.2020.137772.
Minor editing of English is required.
Reviewer 2 Report
Rachid Muleia (ijerph-2452677) presented an informative study assessing the climate change on the national health sector, which is new to the country studied and also offer a baseline for future studies in the field. I have only few concerns which may improve the manuscript.
1. “2.1. Study area and climate”,there are too much descriptive information, it is possible to make table to categorize the parameters in a logic manner.
2. I am not sure if figure one is needed, considering the quality is also not good.
3. "2.3.1. Climate data", the original datasets or raw information should be supplemented as the excel tables for this manuscript, this is also important for data transparency.
4. "2.2. Study design", Is it possible that to make the figure for summarizing this condensed text.
5. Same data should be available for "2.3.2. Epidemiological data " and "2.3.3. Socio-economic data"
6. Line 173-192, please go directly to what and how you did the analysis, the explanation should be in the results or discussion.
7. Line 343, where is the figure A1, I only see figure 1,2,3,4...
8. "5. Conclusion", this part should be a paragraph with 6-10 sentences with emphasizing the major 3-4 major key points, which this study would like to provide.
9. The impact of health, and health index is not clear, what does it contains, you may need a paragraph to explain what exactly it is, that is to say, you also need more refs information for those studied with the consequences of CC on health.
good language to follow.
Reviewer 3 Report
Abstract
- Abbreviations should be avoided in an abstract unless a term is used multiple times.
- Please provide the data analysis.
Keywords
- Ensure keywords does not repeat part of the manuscript title.
Materials and methods
Study area and climate
- Figure 1, please provide the map scale and direction.
Data source
- Page 4, Epidemiological data, why collected data for two years and focus on malaria, diarrhea, tuberculosis, HIV, and chronic and acute malnutrition? Please provide the International Statistical Classification of Diseases and Related Health Problems (ICD).
- Page 7-8, Table 1, please provide the year for each indicator.
Results
- Page 11, Figure 4, please check the figure captions or legends for each figure.
- Page 12, Global health vulnerability index, what is global health? Please clarify.
- Page 13, Figure 5, please check the figure captions or legends for each figure.
- Page 14, Figure 6, why 161 districts that different from abstract and results sections?
Discussion
- Please discuss about the Health National Adaptation Plan (HNAP), National Adaptation Programme of Action (NAPA), and others.
- Please provide the ethical considerations.
Reviewer 4 Report
This paper investigates the relations the country’s areas vulnerability and adaptability for climate change and how the past CC components affect the health sector. It uses a set of indicators that for various components describe the whole of the country and identify weaknesses as well as highly vulnerable regions. In general, it is well written, and the results are numerous but constructively displayed.
Some language editing is needed (see below), and some points are missing in the discussion: (1) the choice for the vulnerability assessment model for which Lines 173/203 can be used (2) It is desirable to include a guideline paragraph on creation of policies: many vulnerable regions are mentioned in the discussion, but a wrap-up giving input how to build constructive guidelines building resilience, where there is need for focus. For instance, the Lines 509-528 should be repositioned in the discussion for that purpose.
Please address the following issues in your re-submmitted paper:
Note: these 162 regions should be numbered in the map, and an index with region names should be given, perhaps as a supplementary Figure, in order to make the region names of the results section 3 traceable!
Lines 121/124 need to positioned in the discussion (and delete last part): ‘It is noteworthy that the VAA report also included a case study on the situation of two emerging climate-sensitive diseases (dengue and chikungunya) in Mozambique and an assessment of the impact of climate change on Malaria, watery diarrhea, and cholera. Nevertheless, for the current paper, we focused on the analysis of the HVI.
Lines 135 and 139 use spatial resolution units.
Line 138 should read: ‘relative humidity (RH).
Table 1 could be repositioned to start at end of section 2.3.3
Notes to Table 1:
It should become more clear how many indicators per component are listed. So please make sure the following are attended to:
- The column of the ‘components could be more narrow.
- Widen Column ‘Indicators’, in order to have most of the indicator description on one line.
- Separate the indicators in each cell with a wider spacing.
- Start each indicator descriptor with a capital.
- Some spaces are missing after the references.
- Second data source in a cell could be positioned higher.
Line 166: ‘effect of scale’ means differences in surface area of the districts? Please, rephrase!
Lines 173/203 should be repositioned in the introduction or discussion. Certainly not in the methods.
Lines 283/284 should be resphrased: ‘In contrast, for the adaptive capacity map, the yellow and the dark red represent the highest and lowest adaptive capacity, with purple representing the presence of a critical situation.’
Line 296: ‘very what’? Poor?
Lines 289/299 should be rephrased: ‘Moreover, the map shows that the AC is higher in urban areas, which coincides with the capital provinces and contrast with rural areas.
Lines 315/316 should read: ‘Figure A1 (a) and (b) map the access to health services index. In this figure, the mapped values consider the districts (Figure A1 (a)) or exclude the districts with extreme values ( Figure A1 (b)), indicated as gray areas.’
Lines 316 should read: ‘When we consider all districts it is observed that only ….’
Lines 322/323 should read: ‘Furthermore, some clustering of districts with very low access to health can be observed: some located’
Lines 332/333 should read: ‘Figure A1 (c) and (d)’.
Lines 343 should read: ‘Figure A1 (e) and (f) show’
Should read: ‘Figure A1. Adaptive capacity determinants maps. Map A and B show the distribution of the health access index with and without outliers, respectively. Map C and D show the distribution of the human resource index with and without outliers, respectively. In Map E and F, we display the water and sanitation index with outliers and discarding outliers, respectively. Note that the regions identified as outliers in B, D and F are depicted as gray areas.’
Actually, I think Figure A1 should be taken up in the core of the manuscript and numbered accordingly.
Line 356 should read: ‘Figure 5 shows the districts’ overall health vulnerability index maps.’
Line 368 should read: ‘index—, the majority of the districts fall into moderate HVI. Moreover, it is observed that…’
Should read: ‘Figure 5. Maps of health vulnerability index by districts. Map (a) presents the global health vulnerability index (GHVI) without discarding extreme values, and Map (b) shows the GHVI discarding extreme values indicated as gray areas’.
Lines 372/373 should read ‘In the current study, we also computed the HVI for each extreme events, droughts, floods, and cyclones.’
Line 374 should read: ‘The HVI to particular, extreme events of droughts, floods and cyclones are presented in Figure 6 (a), (b) and (c) respectively’.
Line 411: replace ‘This’ with ‘The’
Line 422: replace ‘suggested’ with ‘suggest’
Line 425/426 should read: ‘Nevertheless, a more in-depth analysis (not shown herein) suggest that although the country is prone to extreme weather events, few cholera outbreaks were experienced.’
Discussion is missing (1) the choice for the vulnerability assessment model for which Lines 173/203 can be used (2) It is desirable to include a guideline paragraph on creation of policies: many vulnerable regions are mentioned in the discussion, but a wrap-up giving input how to build constructive guidelines building resilience, where there is need for focus. For instance, the Lines 509-528 should be repositioned in the discussion for that purpose.
Lines 496 should read: ‘have the potential to influence’
Abstract
Lines 6/10 should read: ‘Following a methodology proposed by the World Health Organization and the Intergovernmental Panel for Climate Change, a six-step vulnerability and adaptation assessment was conducted to conduct the Health Vulnerability Index (HVI) for Mozambique’s regions (n=162). The HVI integrates historical climate epidemiological, and socio-economic data at the district level, and was computed using exposure, sensitivity, and adaptive capacity dimensions. The results revealed...’
Line 27/28 should read: ‘climate information, financial, human, and natural resources, as well as past and on-going armed conflicts [3]’. Also, replace the reference here! Not correct!
Line 21 should read: ‘domestic product will decrease by 3%,’
Line 38 should read: ‘will therefore be dispersed by’
Line 46: replace ‘climate change’ with CC
Lines 49/50 proposal: ‘In the last 42 years, the country has registered 15 droughts, 20 floods and 26 tropical cyclones [10], with these events often occurring consecutively within the same year. For example, in March 2019, ….’ and ‘Line 57: ‘Subsequently, in less than two months after the cholera outbreak, the country was again….’
Line 57: replace ‘two thousand’ with 2,000
Line 60: replace ‘18 thousand’ with 18,000
Line 67/69: should read: ‘To the best of our knowledge, this is the first study to assess the vulnerability of Mozambique’s health sector to climate change on a national scale.’
Methods:
Lines 72/73 should read: ‘along the inter-tropic convergence zone responsible for Southern Africa’s rainfall patterns.’
Subscript to figure 1 should read: ‘Figure 1. Map of the study area. On the left is the African continent, and marked with a red square is the study area of which the enlargement shows Mozambique and its 162 regions.’
Note: these 162 regions should be numbered in the map, and an index with region names should be given, perhaps as a supplementary Figure, in order to make the region names of the results section 3 traceable!
Line 90: Start a new paragraph after reference 19. New paragraphs starts with ‘Furthermore,…
Lines 100/104 should read: ‘As recommended by the IPCC and the vulnerability sourcebook, the VAA should be conduced as an integrated approach, and for that the present VAA took into account the climate-sensitive health aspects, social, economic and environmental aspects of vulnerability [22]. In accordance with WHO [21], and recommendations by Fritzsche et al. [22], the VAA was…’
Lines 106/121 should read: 1) define the scope of the VAA — at this stage, the objectives of the VAA were specified, the geographic area (for this assessment it was decided to work at district level for the whole country), review of relevant policies, stakeholder engagement for disease/health priorities (this involved workshops, drawing experts from different organizations, including public health disease surveillance, epidemiologist, climatologist, etc.); 2) Identification and selection of indicators — this included defining the necessary data to be collected as well as mapping the existing data sources; 3) Obtaining data —this phase consisted in gathering all relevant data for the VAA (climatological, socio-demographic, health and economic data ), data cleaning and formatting; 4) vulnerability analysis— at this stage the HVI was defined and quantified, preceded by normalization of all the indicators; 5) results validation — the results validation process was done by discussing the computed HVI with Ministry of Health, different stakeholders and experts; and the sixth step involved the dissemination of the results. Additionally, the HVI computed was compared using the approach defined herein to computations through principal component analysis. The VAA ended with a final report that included recommendations for health sector adaptation and climate change monitoring and technical presentation to stakeholders.
Lines 121/124 need to positioned in the discussion (and delete last part): ‘It is noteworthy that the VAA report also included a case study on the situation of two emerging climate-sensitive diseases (dengue and chikungunya) in Mozambique and an assessment of the impact of climate change on Malaria, watery diarrhea, and cholera. Nevertheless, for the current paper, we focused on the analysis of the HVI.
Lines 135 and 139 use spatial resolution units.
Line 138 should read: ‘relative humidity (RH).
Line 149 omit ‘also’.
Line 151/152 should read: ‘To compute HVI, epidemiological data on annual cases of malaria, diarrhea, tuberculosis, HIV, and chronic and acute malnutrition, were collected.’
Lines 154/155 should read: ‘For the current study, socio-economic data were considered that were collected from various sources: the Ministry of Health (MISAU), the Technical Secretariat for Food Security and Nutrition (SETSAN), and the National Institute of Statistics (INE). From the latter, demographic data, literacy rate, per capita expenditure and the rate of accessibilityto water, sanitation and hygiene (WASH), were obtained.
Table 1 could be repositioned to start at end of section 2.3.3
Notes to Table 1:
It should become more clear how many indicators per component are listed. So please make sure the following are attended to:
- The column of the ‘components could be more narrow.
- Widen Column ‘Indicators’, in order to have most of the indicator description on one line.
- Separate the indicators in each cell with a wider spacing.
- Start each indicator descriptor with a capital.
- Some spaces are missing after the references.
- Second data source in a cell could be positioned higher.
Line 166: ‘effect of scale’ means differences in surface area of the districts? Please, rephrase!
Lines 173/203 should be repositioned in the introduction or discussion. Certainly not in the methods.
Lines 194/195 should read: ‘To assess the vulnerability, the framework recommended by the IPCC and WHO was considered, whereby…’
Line 196 should read: ‘namely: ‘
Line 210/211should read ‘For the purpose of this study, equal weights for all the normalized indicators were considered, which…’
Line 225 should read: ‘All the analyses herein were performed using’
‘Figure 2. Spatial distribution of climate variables. Figure (a) shows the spatial distribution of average temperature. (b) shows the spatial distribution of average rainfall and (c) shows the spatial distribution of relative humidity, for the period between 1979 and 2016.’
Line 267/268 should read: ‘Figure 3 shows the incidences of extreme events—cyclones, droughts, and floods—in Mozambique for each of the districts in the last 40 years.’
Line 261/262 should read: ‘are more frequent in the Nampula province’
Line 263 start a new paragraph on ‘Droughts’, this starting with ‘Figure 3 (b) shows.
Figure 3. Spatial distribution of extreme events. Maps (a), (b) and (c) show the incidences of cyclones, droughts and floods for each district across the country, respectively. Extreme events are reported between 1979 and 2019.
Section 3.2
Lines 283/284 should be resphrased: ‘In contrast, for the adaptive capacity map, the yellow and the dark red represent the highest and lowest adaptive capacity, with purple representing the presence of a critical situation.’
Line 296: ‘very what’? Poor?
Lines 289/299 should be rephrased: ‘Moreover, the map shows that the AC is higher in urban areas, which coincides with the capital provinces and contrast with rural areas.
Line 308 use: ‘contribute’
Lines 315/316 should read: ‘Figure A1 (a) and (b) map the access to health services index. In this figure, the mapped values consider the districts (Figure A1 (a)) or exclude the districts with extreme values ( Figure A1 (b)), indicated as gray areas.’
Lines 316 should read: ‘When we consider all districts it is observed that only ….’
Lines 322/323 should read: ‘Furthermore, some clustering of districts with very low access to health can be observed: some located’
Lines 332/333 should read: ‘Figure A1 (c) and (d)’.
Lines 343 should read: ‘Figure A1 (e) and (f) show’
Should read: ‘Figure A1. Adaptive capacity determinants maps. Map A and B show the distribution of the health access index with and without outliers, respectively. Map C and D show the distribution of the human resource index with and without outliers, respectively. In Map E and F, we display the water and sanitation index with outliers and discarding outliers, respectively. Note that the regions identified as outliers in B, D and F are depicted as gray areas.’
Actually, I think Figure A1 should be taken up in the core of the manuscript and numbered accordingly.
Line 356 should read: ‘Figure 5 shows the districts’ overall health vulnerability index maps.’
Line 368 should read: ‘index—, the majority of the districts fall into moderate HVI. Moreover, it is observed that…’
Should read: ‘Figure 5. Maps of health vulnerability index by districts. Map (a) presents the global health vulnerability index (GHVI) without discarding extreme values, and Map (b) shows the GHVI discarding extreme values indicated as gray areas’.
Lines 372/373 should read ‘In the current study, we also computed the HVI for each extreme events, droughts, floods, and cyclones.’
Line 374 should read: ‘The HVI to particular, extreme events of droughts, floods and cyclones are presented in Figure 6 (a), (b) and (c) respectively’.
Line 411: replace ‘This’ with ‘The’
Line 422: replace ‘suggested’ with ‘suggest’
Line 425/426 should read: ‘Nevertheless, a more in-depth analysis (not shown herein) suggest that although the country is prone to extreme weather events, few cholera outbreaks were experienced.’
Discussion is missing (1) the choice for the vulnerability assessment model for which Lines 173/203 can be used (2) It is desirable to include a guideline paragraph on creation of policies: many vulnerable regions are mentioned in the discussion, but a wrap-up giving input how to build constructive guidelines building resilience, where there is need for focus. For instance, the Lines 509-528 should be repositioned in the discussion for that purpose.
Lines 496 should read: ‘have the potential to influence’
Round 2
Reviewer 4 Report
All previous suggestions have been satisfactory dealt with, making the paper acceptable for publication.
Except for 'Note: these 162 regions should be numbered in the map, and an index with region names should be given, perhaps as a supplementary Figure, in order to make the region names of the results section 3 traceable!' Basically, it was meant to provide a map of Mozambique with all the 162 regions numbered, and additionally have the names of the regions listed corresponding to the numbers in the map, just to follow more clearly the use of region names in the results and discussion section. It is clearly understandable that putting the names in the regions of the map would not work well. If the authors can find time to prepare such a map as a supplementary Figure, that would enhance the readability of the paper.
